# Impaired Alveolar Re-Epithelialization in Pulmonary Emphysema

**DOI:** 10.3390/cells11132055

**Published:** 2022-06-28

**Authors:** Chih-Ru Lin, Karim Bahmed, Beata Kosmider

**Affiliations:** 1Department of Microbiology, Immunology and Inflammation, Temple University, Philadelphia, PA 19140, USA; chih-ru.lin@temple.edu; 2Center for Inflammation and Lung Research, Temple University, Philadelphia, PA 19140, USA; karim.bahmed@temple.edu; 3Department of Thoracic Medicine and Surgery, Temple University, Philadelphia, PA 19140, USA

**Keywords:** alveolar epithelium, alveolar type II cells, emphysema, tissue homeostasis, regeneration, lung repair

## Abstract

Alveolar type II (ATII) cells are progenitors in alveoli and can repair the alveolar epithelium after injury. They are intertwined with the microenvironment for alveolar epithelial cell homeostasis and re-epithelialization. A variety of ATII cell niches, transcription factors, mediators, and signaling pathways constitute a specific environment to regulate ATII cell function. Particularly, WNT/β-catenin, YAP/TAZ, NOTCH, TGF-β, and P53 signaling pathways are dynamically involved in ATII cell proliferation and differentiation, although there are still plenty of unknowns regarding the mechanism. However, an imbalance of alveolar cell death and proliferation was observed in patients with pulmonary emphysema, contributing to alveolar wall destruction and impaired gas exchange. Cigarette smoking causes oxidative stress and is the primary cause of this disease development. Aberrant inflammatory and oxidative stress responses result in loss of cell homeostasis and ATII cell dysfunction in emphysema. Here, we discuss the current understanding of alveolar re-epithelialization and altered reparative responses in the pathophysiology of this disease. Current therapeutics and emerging treatments, including cell therapies in clinical trials, are addressed as well.

## 1. Introduction

Pulmonary emphysema, a form of chronic obstructive pulmonary disease (COPD), is characterized by alveolar septa destruction and airflow limitation [1]. Alveoli are progressively damaged in this disease, mainly due to exposure to cigarette smoke and environmental irritants, resulting in oxidative stress and high inflammation. Diagnosis is based on lung structural abnormalities on chest computed tomography (CT), respiratory alterations, and spirometric measures. There are limited effective treatment strategies available to manage symptoms and prevent exacerbations. For patients with advanced disease, lung volume reduction surgery and lung transplants are considered. Alveolar type II (ATII) cells have stem cell potential to maintain alveolar epithelial homeostasis and promote re-epithelialization after injury [2,3]. Various signaling pathways precisely regulate ATII cell proliferation and differentiation [4]. Additionally, their microenvironment, which includes immune cells, mesenchyme, and endothelial cells, plays a critical role. However, alveolar epithelial cell injury, exhaustion, and insufficient re-epithelialization disturb homeostasis and contribute to structural damage in emphysema. An aberrant repair can rebound adversely on injury and lead to lung destruction. There are some cell therapies for patients with emphysema in clinical trials. Nevertheless, the mechanism of alveolar re-epithelialization is still elusive, and treatments targeting the signaling for ATII cell proliferation and differentiation have not been developed. It has been challenging to find therapeutic targets and strategies to stimulate alveolar re-epithelialization and lung regeneration due to the histological complexity of the lung and cell types involved. A better understanding of lung development and alveolar regenerative processes may lead to a breakthrough in emphysema treatment.

## 2. The Role of the Alveolar Epithelium

The mammalian lung is a complex organ with a highly branched network [5]. The trachea divides into two main bronchi that extend toward the bronchial tree. The distal regions of the bronchial tree transform during development into densely packed alveolar sacs that mediate the gas exchange. The alveolar epithelium consists of alveolar type I (ATI) cells and ATII cells [2]. ATI cells are large squamous cells that cover 95% of the alveolar surface and participate in gas exchange. Podoplanin (T1α), a receptor for advanced glycation end-products (RAGE), and aquaporin 5 (AQP5) are commonly used as ATI cell markers. ATII cells have a cuboidal shape, lamellar bodies, and microvilli. They produce and secrete pulmonary surfactants to reduce the lungs’ surface tension and maintain alveolar fluid homeostasis. Pulmonary surfactant is composed of lipids and proteins that form a thin layer of fluid coating the alveolus at the air–liquid interface. Surfactant proteins (SP)-B and SP-C are small and hydrophobic proteins, lowering the surface tension in the alveoli. SP-C and its precursor pro-SP-C are used as specific markers of ATII cells. SP-A and SP-D are hydrophilic proteins that have important roles in host innate immunity. ATII cells are more resistant to injury than ATI cells and are essential for the alveolar microenvironment, innate immunity, and re-epithelialization of the damaged lung [2]. The alveolar epithelium is surrounded by the capillary endothelium covered with pericytes and extracellular matrix (ECM). The integrity of the alveolar–capillary barrier is important for ATII cell function and homeostasis. Various cell types constitute a specific microenvironment to support alveolar repair after injury (Figure 1).

## 3. ATII Cell Regulation in Alveolar Re-Epithelialization

Post-pneumonectomy, respiratory infections, and idiopathic pulmonary fibrosis (IPF) animal models are commonly used to study the alveolar injury and repair systems. IPF is a fatal pulmonary disease with abnormal scarring of connective tissue, resulting from increased proliferation of fibroblasts around alveoli and excessive ECM deposition. It was demonstrated that impaired alveolar regeneration causes sustained ATII cell exposure to elevated mechanical tension, which activates the TGF-β signaling loop in ATII cells and promotes fibrosis progression [6]. However, fibroblasts from individuals with emphysema manifested a less reparative response [7,8], indicating the damaged lung microenvironment in emphysema. Despite clinical, radiological, and pathological differences, emphysema and IPF develop in elderly people and share similarities [9]. They are related to long-term exposure to environmental irritants, mainly cigarette smoking. Both diseases are characterized by progressive loss of alveolar parenchyma, leading to airflow limitation. Dysregulated ATII cell function is observed in emphysema patients, although there is limited research on their regenerative role. Of note, combined pulmonary fibrosis and emphysema (CPFE) is clinically defined by the coexistence of upper lobe emphysema with lower lobe fibrosis, whereas its mechanisms remain largely unknown. The complicatedness of signaling and microenvironmental interactions, individual susceptibility, genetic background, and gene regulation might account for the divergent consequences of emphysema development.

It has been shown that alveolar cell turnover is low at a steady state [3,10]. Lineage-labeled ATII cells remained constant during long-term homeostasis with ATI replacement intermittently for over 16 months in mice. Nonetheless, apparent regeneration after lung injury has been reported in animals and humans [11,12,13]. ATII cells play a critical role in adult lungs [14,15]. They have stem cell properties and are capable of self-renewal and differentiation into ATI cells to restore the structure and function of the alveoli in both homeostatic and regenerative states. This was determined by single-cell sequencing, lineage-tracing studies, various lung injury models, and 3D organoid culture in mice [3,5,10,16]. SP-C^+^ ATII cells proliferated and differentiated into ATI cells to repair alveolar epithelium by about 21 days after diphtheria toxin A (DTA)-targeted injury [3]. Of note, SCGB1A1^+^ secretory cells at the bronchoalveolar duct junction (BADJ) helped to restore the alveolar epithelium after severe injury. This indicates the role of other progenitors in alveolar repair. Importantly, the alveolosphere formation was observed in human ATII cells cultured with fibroblasts [3,15]. Studies further indicate the essential role of mesenchymal cells as the ATII cell niche. Various factors and signaling pathways have been demonstrated in ATII cell proliferation and differentiation (Figure 2). They are dynamically regulated to maintain ATII cell function. However, more evidence using human samples is required.

### 3.1. Mechanical Forces

The unique mechanical properties of lungs build a complex and dynamic relationship between lung cells and their microenvironment. It has been shown that mechanical forces from amniotic fluid inhalation in fetal mice regulate the differentiation of ID2^+^ alveolar progenitor cells during alveolar development, especially high mechanical tension is required for their differentiation to ATI cells [17]. By using a time-lapse imaging system, an actin-based cell protrusion process was observed from the tip of distal airways toward the mesenchyme before the alveolar progenitor cell differentiation. Specifically, fibroblast growth factor (FGF)10/FGF receptor (FGFR)2-mediated ERK1/2 signaling from mesenchymal cells controls the alveolar progenitor cell protrusion. This ensures that these protruded alveolar progenitors become cuboidal ATII cells during distal sac formation. In contrast, non-protruded cells become flat ATI cells. The myosin enrichment at the apical region of protruded cells provides structural support and prevents ATII cells from being flattened by mechanical forces. Furthermore, mechanical stress has been reported as an important factor that initiates alveolar regeneration from post-pneumonectomy in humans and animals. A case report describes new lung growth in a patient after a right-sided pneumonectomy to treat lung adenocarcinoma [11]. The 15-year follow-up observation shows a progressive increase in the forced expiratory volume in 1 s (FEV1), forced vital capacity (FVC), tissue volume, and alveolar number, although with a shallower alveolar structure compared to normal lungs. The patient’s relatively young age and daily exercise program are reminiscent of the role of stretching in lung development. Specifically, this suggests that cyclic stretch might be an important trigger for new lung growth. Likewise, loss of alveoli after pneumonectomy in mice increases the mechanical tension of the alveolar epithelium [18]. It motivates actin polymerization in ATII cells and subsequent YAP-dependent cell proliferation. Fewer ATII cell proliferation and differentiation to ATI cells were observed in the lungs with an inserted prosthesis compared to pneumonectomy lungs. This highlights the importance of mechanical forces in alveolar re-epithelialization and implies the role of adherens junction and tight junction structures in regulating ATII cell response.

Under pathological conditions with altered molecular levels, mechanical forces are prone to cause stress failure of the remodeled alveolar wall, contributing to the progression of emphysema. It was demonstrated that elastase-treated rat lungs had a lower threshold for mechanical failure in vitro [19]. The newly deposited elastin–collagen networks in emphysematous lungs significantly exhibited distortion/deformation during mechanical stretching. Additionally, resistive breathing by tracheal banding in mice induced pulmonary inflammation and exacerbations of elastase-induced emphysema [20]. This points out the role of elastic fiber damage and abnormal mechanical environment in developing alveolar wall stress. It further emphasizes the interconnectedness of multilevel changes in this disease and might explain the limited efficacy of current therapeutics.

### 3.2. WNT/β-Catenin Signaling

WNT signaling is a central mechanism regulating tissue morphogenesis and repair in various organs [21]. It controls the development of the embryonic and fetal lung, homeostasis, and regeneration after injury [22]. It is activated by the interaction of WNT ligands and Frizzled (FZD) receptors, leading to a canonical or non-canonical/β-catenin-independent pathway. WNT ligands are a large family of secreted and cysteine-rich glycolipoproteins with 19 members in humans, implying their complexity of signaling regulation and biological output in stem/progenitor and niche cells in a region-dependent manner during regeneration.

A small subset of ATII cells with AXIN2 expression was identified that maintains their homeostatic turnover in a WNT/β-catenin-dependent manner in humans and mice [15,16]. They promote murine ATII cell growth in ex vivo organoids. Activation of WNT signaling increases the expansion of ATII cells, whereas inhibition of this signaling inhibits ATII cell growth and favors their differentiation towards ATI cells [23]. Lineage tracing experiments demonstrated that these WNT-responsive AXIN2^+^ ATII cells significantly proliferate and differentiate into ATI cells at injured alveoli caused by H1N1 influenza virus infection in mice. In murine lungs, adjacent PDGFRα^+^ fibroblasts are the source of secreted WNTs. ATII cells further induce autocrine WNTs to expand the progenitor pool after injury. Pharmacological or genetic inhibition of WNT signaling reduced the pool [15]. Neutrophil transmigration also activates WNT signaling in human ATII cells for repair [24]. In humans, TM4SF1^+^HTII-280^+^EpCAM^+^ ATII cells form many larger organoids containing ATII and ATI cells [15]. These ATII cells are also responsive to WNT signaling, and its activation promoted ATII cell formation while inhibition of this pathway induced ATII cell differentiation into ATI cells. RNA sequencing analysis demonstrates that they are evolutionarily conserved with mouse AXIN2^+^ ATII cells, and they both are enriched for WNT signaling targets, including AXIN2 and FGFR2, the primary receptor for FGF7 and FGF10. FGF7 or FGF10 treatment of these WNT-responsive adult murine or human ATII cells increased colony size and colony-forming efficiency in organoid cultures [15]. Skronska-Wasek et al. further demonstrated that FZD4 promoted the canonical WNT/β-catenin signaling and ATII cell function for lung repair and elastogenesis in mice [25]. These findings suggest that the WNT/β-catenin pathway is crucial for promoting ATII cell expansion and function in alveolar re-epithelialization.

### 3.3. BMP/SMAD Signaling

BMPs and related antagonists are a group of signal molecules involved in TGF-β signaling in multiple organ systems, including the lung [26]. They are essential in embryogenesis, development, and tissue homeostasis, such as the regeneration of tracheal epithelium from basal stem cells [27], fracture repair, and vascular remodeling [26]. They are highly conserved across many species. Mutations of BMPs result in embryonic lethality or severe defects in mice and human diseases. *Tgfbr2*, *Smad3*, and *Itgb6* knockout mice displayed alveolar destruction [28], suggesting the predisposition of insufficient TGF-β signaling to lung diseases. It was demonstrated that BMP signaling is active in the alveolar epithelium homeostasis in mice [29]. *Bmp6*, *Bmp2*, *Bmpr1a*, and *Bmpr2* were significantly reduced following pneumonectomy. BMP4 inhibits ATII cell proliferation and induces differentiation into ATI cells in the pneumonectomy-induced lung repair mouse model and alveolar organoids. Studies of gain or loss of BMP signaling in vivo indicate that its dysregulation attenuated the self-renewal of murine ATII cells [29]. In support, single-cell RNA sequencing on murine ATII cells following lipopolysaccharide (LPS)-induced lung injury revealed distinct subpopulations of proliferating, cell cycle arrest, and differentiating ATII cells [30]. The genome-wide expression analysis indicated that upregulation of TGF-β signaling correlated with cell cycle arrest, leading to inhibition of ATII cell proliferation. This pathway was downregulated in ATII cells as differentiating to ATI cells. Moreover, a recent study showed the activation of TGF-β signaling in response to mechanical tension post-pneumonectomy in mice [6]. Loss of CDC42 function, belonging to the RhoGTPase family essential for actin polymerization, in ATII cells impaired their differentiation into ATI cells and alveolar regeneration post pneumonectomy. These cells were continuously exposed to elevated mechanical tension that activated TGF-β signaling in ATII cells, resulting in a sustained ATII-ATI cell transitional state. ATII cell function during alveolar re-epithelialization appears to be tightly regulated by TGF-β/BMP stimuli and antagonism.

### 3.4. NOTCH Signaling

NOTCH signaling is a highly conserved cell–cell communication mechanism through direct receptor–ligand interaction that regulates embryonic development and tissue homeostasis in adults [31]. It was activated in murine ATII cells in *Pseudomonas aeruginosa*-induced acute lung injury [32]. DLK1-mediated NOTCH inhibition was temporarily elevated during repair, which is required for ATII cell to ATI cell differentiation. Specific *Dlk1* disruption in ATII cells did not affect their proliferation, yet it caused persistent NOTCH activation and impaired ATII-to-ATI cell transition, resulting in the accumulation of an intermediate cell population with low levels of both ATII and ATI cell markers. This shows the precisely timed regulation of NOTCH signaling for alveolar repair [32]. The hyperactive NOTCH signaling in human lungs and ATII cells was found in progressive fibrotic lung diseases, indicating that a high-to-low NOTCH switch is essential for ATII cell differentiation [33,34]. In particular, NOTCH2 activation in murine ATII cells contributes to alveolar formation in the developing lung [35]. *Notch2* disruption resulted in abnormal enlargement of the alveolar spaces. NOTCH2 regulates epithelial-mesenchymal interactions through PDGF-A ligand induction and subsequent paracrine activation of PDGFRα signaling in alveolar myofibroblasts. It also maintains the integrity of the epithelial and smooth muscle layers of the distal conducting airways [35]. In rat fetal ATII cells, the addition of DLK1 increased mRNA and protein expression of SP-C, NOTCH1, and HES1 and promoted ATII cell proliferation with low differentiation. In response to NF-κB, ATII cells increased canonical NOTCH ligand JAG1 and down-regulated SP-C levels, leading to accelerated ATII-ATI cell differentiation. This recapitulates the ATII cell regulation by different NOTCH ligands at the homeostatic and inflammatory phases [36].

### 3.5. YAP/TAZ Signaling

YAP and TAZ are key transcriptional coactivators in HIPPO signaling and are important in organogenesis, tissue homeostasis, and regeneration [37]. ATII cell-specific *Taz* conditional knockout mice showed abnormal alveolar development, emphysema-like changes, and increased inflammatory cells in bronchoalveolar lavage fluid (BALF) while *Yap* knockout mice are embryonic lethal [38]. The importance of nuclear TAZ was demonstrated during ATII cell differentiation into ATI cells using ATII cell-specific *Taz* deletion in the organoid culture system and bleomycin-induced lung injury model [39]. Further, CDC42-controlled JNK and p38 MAPK activation mediated the nuclear YAP expression in ATII cells for alveolar regeneration in post pneumonectomy murine lungs [18]. Similarly, YAP/TAZ expression was required in ATII cell proliferation and differentiation into ATI cells in mice infected with *Streptococcus pneumoniae* [40]. ATII cell-specific *Yap*/*Taz* knockout mice displayed prolonged inflammation and delayed alveolar re-epithelialization, accompanied by the dysregulation of *Ikba* and *Ikbb*. This contributed to the development of severe fibrotic lesions within the alveoli. *Yap/Taz* deficiency also reduced *Fgf1*, *Fgfr3*, *Wnt3a*, *Bmp4*, *Tgfb2*, and *Egfr* levels in murine SP-C^+^ cells. Of note, these genes are positively correlated with ATII cell proliferation and differentiation under normal conditions, therefore suggesting a complex interplay of these signaling pathways and their role in the maintenance of alveolar epithelial integrity.

### 3.6. ATII to ATI Cell Transitional State

During alveolar regeneration, ATII cells can give rise to ATI cells following a stepwise transition of different cellular states [41,42,43]. There is limited research about this process and its precise mechanisms. It was reported that the TGF-β level was low in the proliferating murine ATII cells, upregulated in the transitional state, and downregulated in differentiated cells [4]. TGF-β treatment markedly decreased SP-A, proSP-B, and SP-C protein expression, but increased SP-D levels in primary human ATII cells [44]. It also reduced *FAS* mRNA levels and the phospholipid-transporting ATPase ABCA3, important for surfactant production and transportation. Besides, active WNT pathways (e.g., increased WNT ligands *WNT4* and *WNT7a* and receptors *FZD1*, *FZD2*, and *FZD7*) and downregulated *WNT5a* and *SP-C* mRNA levels were detected during normal human ATII cell to ATI cell differentiation using Affymetrix microarray [45]. Significantly, WNT7a upregulated AQP5 mRNA and protein expression. Recently, Kobayashi et al. identified a CLDN4^+^KRT19^+^SFN^+^ population in the transitional state between ATII and ATI cells using organoid cultures, single-cell transcriptome studies, lineage tracing, and lung injury models induced by bleomycin or LPS in mice [41]. Pathway enrichment analysis revealed that P53, TGF-β signaling, TNF–NF-κB, ErbB, hypoxia-inducible factor (HIF1), HIPPO–YAP, cell-cycle arrest, cytoskeletal dynamics, tight-junction, cellular senescence, and DNA damage response (DDR) are activated. Particularly, P53 directly controls the transcriptional gene regulation in this state. Elevated KRT8 levels were also found in this transitional state during alveolar regeneration in bleomycin-induced murine lung injury [42,43]. These studies indicate numerous molecular regulation that contributes to ATII cell morphological and functional changes during alveolar re-epithelialization.

### 3.7. Dysregulated Alveolar Re-Epithelialization in Emphysema

Oxidative stress and high inflammation trigger a series of dysregulated signaling pathways and cell dysfunction, leading to loss of cell homeostasis and alveolar destruction in emphysema [46]. Alveolar epithelial cell apoptosis/proliferation imbalance was reported in patients with this disease [47]. In particular, reduced numbers of nuclear β-catenin-positive ATII cells were displayed in emphysema [48], indicating WNT-responsive ATII cells’ dysfunction in the re-epithelialization. Similarly, this signaling activity was down-regulated in mouse models of emphysema [25,48]. It was further revealed that cigarette smoke attenuated the WNT receptor FZD4 protein expression in vivo and in vitro [25]. Reduced *FZD4* gene expression was observed in ATII cells in emphysema patients and the lungs in mouse models of this disease induced by exposure to cigarette smoke or elastase. WNT/β-catenin activation by lithium chloride attenuated parenchymal destruction and restored alveolar structure in elastase-challenged mice [48]. Besides, WNT/β-catenin activity is required for alveolar organoid formation from murine ATII cells [49]. However, those ATII cells isolated from elastase-induced emphysema had a low capacity to form organoids. Aged murine ATII cells also showed low progenitor potential in organoid cultures [50]. These studies suggest dysregulated WNT signaling for alveolar re-epithelialization. Nonetheless, chronic activation of canonical WNT/β-catenin signaling induced ATII cell senescence with decreased SP-C expression leading to profibrotic changes [50], indicating their contribution to a fibrotic environment. Additionally, the imbalance between canonical and non-canonical WNT signaling has been demonstrated in aging-related diseases, including emphysema. Specifically, non-canonical WNT ligands WNT-5A and WNT-5B were increased in patients with this disease [51,52], the lungs of aged mice [53], and experimental mouse emphysema models [54]. Functionally, they suppressed the growth and differentiation of ATII cells through negative regulation of the canonical WNT signaling pathway, thereby contributing to lung disease development and progression [52,53].

There is evidence of high collagen deposition in emphysema [55,56], indicating the excessive ECM degradation and remodeling of alveoli. Aberrant repair after alveolar epithelial cell damage can backfire and lead to lung destruction. It has been shown that cigarette smoke inhibits fibroblastic functions related to alveolar regeneration in vitro [57]. The patient-derived lung fibroblasts were less responsive to TGF-β and had reduced repair activity, as determined by chemotaxis and collagen gel contraction assays [7]. It was accompanied by increased prostaglandin E (PGE) and the receptors EP2 and EP4 protein expression. In support, emphysema-derived mesenchymal stromal cells expressed lower *FGF-10* and hepatocyte growth factor (*HGF*) mRNA and HGF and decorin protein [8], indicating insufficient microenvironmental support for tissue regeneration. TGF-β represents a potent mediator of lung diseases, including emphysema and fibrosis [37]. Besides ATII cell proliferation and differentiation, TGF-β regulates ECM components and tissue remodeling. It senses and modulates mechanical stress generated from ECM rigidity and cell density. Specifically, *TGFB2* is associated with emphysema by genome-wide association studies (GWAS) in humans [58,59], highlighting its physiopathological role in the lung and the potential clinical significance in expediting regeneration. Additionally, HHIP is a genetic locus linked to emphysema susceptibility in humans and mice [60]. It is a component of hedgehog signaling required for myofibroblast differentiation and mesenchymal proliferation. Aberrant hedgehog activation in the distal fibroblasts disrupts mesenchymal identity and ATII cell niche, leading to emphysematous changes in mice [61]. Hedgehog-receptive mesenchyme is conserved between mice and humans as detected by single-cell analysis [62]. These suggest the disrupted alveolar environment and imbalance between lung injury and repair in emphysema (Figure 3).

Recently, it was demonstrated that endothelium dysfunction increased the severity of emphysematous changes in patients and elastase-treated mice [63]. Delivery of healthy lung endothelial cells reversed alveolar wall destruction, improved lung function, and increased lung elastance in the elastase-induced mouse model of emphysema, possibly through promoting both endothelial and non-endothelial cell proliferation and tissue remodeling. This not only restates the pathophysiological role of ATII cell niches but also highlights the therapeutic value of endothelial cells for alveolar repair. The dysregulated mechanism of alveolar repair is still poorly understood. Studies on normal lung regeneration stimulation are challenging but can be the key to new therapeutic approaches in emphysema.

## 4. ATII Cell Niche

### 4.1. Mesenchyme

Epithelial-mesenchymal interactions regulate lung development and restoration of alveolar architecture after lung injury [64]. In the presence of fibroblasts, ATII cells can give rise to alveolospheres in mice and humans [3]. In support, fibroblasts-derived keratinocyte growth factor (KGF) and HGF in the conditioned medium promoted DNA synthesis and proliferation of rat primary ATII cells [65]. Lately, it has been demonstrated that human ATII cells, but not murine ATII cells, can differentiate into metaplastic KRT5^+^ basal cells in organoid cultures and xenotransplant. The pathological mesenchymal niche modulates this differentiation, which resembles severe lung injuries with loss of ATII cells [66]. These bring out the influence of mesenchymal cells on ATII cell function and fate. Mesenchyme-derived FGF-10 expression is precisely controlled for lung epithelial lineage determination in mice, including ATII and ATI cells [67]. During early lung development, cytoplasmic YAP in SOX9-expressing epithelial progenitors suppressed *Fgf-10* expression as differentiating into bronchial epithelium; however, nuclear YAP and β-catenin signaling in these progenitors subsequently promoted *Wnt7b* expression, mesenchymal FGF-10 production, and alveolar epithelial differentiation. Integrin-linked kinase (ILK), a central component of cell–matrix interactions associated with integrins to mediate ECM signals and many cellular processes, was also shown to regulate the HIPPO pathway leading to alveologenesis in later stages of lung development. Conversely, the cultured transitional human primary ATII cells have been shown to inhibit the mRNA levels of *HGF*, *FGF-7*, *FGF-10*, *FN1*, and TGF-β-stimulated connective tissue growth factor (*CTGF*), type I collagen (*COL1A1*), and α-smooth muscle actin (*ACTA2*) in normal lung fibroblasts. These studies exhibit the intimate regulation between ATII and mesenchymal cells.

It is known that TGF-β1 plays a pivotal role in fibroblast activity, collagen synthesis, and mesenchymal–epithelial interactions [26]. Besides alveolar epithelium, active BMP signaling (pSMAD1/5/8) was shown in murine PDGFRα^+^ and PDGFRb^+^ alveolar stromal cells and EMCN^+^ endothelial cells at a steady state [29]. Interestingly, a similar wave of BMP signaling was observed in PDGFRα^+^ alveolar stromal cells after pneumonectomy. *Bmp6* and *Bmpr2* were reduced while the BMP antagonists follistatin (*Fst*) and follistatin-like 1 (*Fstl1*) were increased in these cells. Enhanced BMP signaling in PDGFRα^+^ cells then inhibited ATII cell proliferation in the regenerating lung. Additionally, TGF-β activation impairs the ability of primary human lung fibroblast to support organoid formation from adult mouse lung epithelial EpCAM^+^ cells [53]. The RNA sequencing analysis of the primary human lung fibroblasts incubated with TGF-β showed altered expression of key WNT signaling pathway components and target genes, as well as growth factors such as *Fgf2*, *Fgf7*, *Hgf*, and *Wnt5a*. Supplement with HGF or FGF7 restored organoid formation in this condition. By contrast, TGF-β-induced myofibroblast differentiation increased the proportion of airway organoids (proSP-C^−^/ACT^+^) in mice. These studies show the delicate regulation and role of TGF-β signaling in different cell types and its interaction with other signaling pathways.

### 4.2. Pulmonary Vasculature

The close contact of the alveoli–capillary structure makes pulmonary capillary endothelial cells an important component of the ATII cell niche. Their proliferation was significantly elevated after pneumonectomy in mice [68]. Particularly, it was mediated by vascular endothelial growth factor receptor (VEGFR) 2 and FGFR1 signaling in endothelial cells, thereby selectively up-regulating MMP14 production and promoting epidermal growth factor receptor (EGFR)-mediated alveolar regeneration. These activated endothelial cells supported epithelial cell expansion and formed alveolar–capillary-like sacs in 3D angiosphere co-cultures. Transplantation of wild-type murine pulmonary capillary endothelial cells into *Vegfr2* and *Fgfr1* deficient mice restored lung alveologenesis and improved lung function. Additionally, endothelial cell-produced HGF promoted ATII cell function for alveolar re-epithelialization in mice challenged by bleomycin or hydrochloric acid [69].

It has been demonstrated that endothelial cells released angiocrine sphingosine-1-phosphate (S1P) to promote alveolar re-epithelialization via S1PR2-YAP signaling in mice infected with *Pseudomonas aeruginosa* [62]. Moreover, Kato et al. revealed that *Yap*/*Taz* deficiency in PDGFRβ^+^ pericytes reduced CD31^+^ endothelial and ATII cell proliferation and decreased capillary formation and lung volume in mice [70]. Loss of *Yap* and *Taz* in these pericytes reduced the expression of several growth factors, including *Angpt1*, *Tgfb2*, *Wnt11*, *Bmp4*, and *Hgf*. This altered the paracrine regulation in murine ATII cells, leading to decreased proliferation and defective alveologenesis [70]. Of note, mice with inducible *Angpt1* mutants in PDGFRβ^+^-expressing cells had a similar phenotype as specific *Yap*/*Taz* deletion, such as alveologenesis defects, reduced capillary formation in the secondary septa, and increased airspace volume. Recently, various subpopulations of endothelial cells which express high or low CAR4 were identified in the regenerating region of alveoli following influenza infection or bleomycin treatment in mice [71,72]. This points out the involvement and heterogeneity of alveolar endothelial cells in alveolar regeneration after injury.

### 4.3. Immune Cells

In response to inflammatory insults, immune cells are activated and recruited to the lung injured regions [73]. They release various inflammatory mediators, thus changing the microenvironment for defense and alveolar repair. Increasing evidence showed interrelationships between ATII cells and inflammatory cells in different pathophysiological conditions. For example, SARS-CoV infection induces a vigorous innate immune response in primary human ATII cells. It markedly increased the mRNA levels of interferon (IFN)–β, IFN–λ/interleukin (IL)-29, IFN-responsive genes, and chemokines that recruit inflammatory cells [74]. Katsura et al. have identified that IL-1 and tumor necrosis factor α (TNFα) enhanced ATII cell proliferation through the NF-κB pathway for alveolar regeneration in organoids and influenza-induced murine lung injury [75]. By contrast, IL-13 disrupts the self-renewal and differentiation of ATII cells in genetic mouse studies and human organoids. RNA sequencing analysis of murine organoids grown in the presence of IL-13 revealed increased expression of bronchiolar markers, suggesting the reprogramming of these cells toward a bronchiolar phenotype [76].

More specifically, alveolar macrophages play an important role in immunosurveillance and lipid surfactant catabolism for lung homeostasis [77]. They interact with various components of pulmonary surfactant, including SP-A and SP-D, in the innate host defense and facilitate the resolution of inflammation [78]. Their remarkable plasticity and highly specialized phenotype are determined by the lung microenvironment [77]. Lechner et al. have identified regenerative myeloid subpopulations by single-cell RNA sequencing, including recruited *Ccr2*^+^ monocytes and *Arginase1*^+^ M2-like macrophages, in the murine lung close to ATII cells, after pneumonectomy [79]. In addition to alveolar macrophages, ATII cells recruit these bone marrow-derived macrophages through the CCL2-CCR2 chemokine axis for optimal alveologenesis, along with IL4RA-expressing leukocytes. Without fibroblasts or endothelial cells, F4/80+ macrophages are able to support pneumosphere formation when co-cultured with ATII cells. The cells expressed markers of both ATII and ATI cells, which indicates the macrophages’ contribution to ATII cell survival and promoting its transitional state. Moreover, group 2 innate lymphoid cells (ILC2) were present post pneumonectomy as the source of IL-13 that provided a Th2-type microenvironment to support M2-like macrophage proliferation [79]. Loss of recruited monocytes or IL4RA signaling, a receptor for cytokines IL-4 and IL-13, led to fewer ATI cells and less efficient lung regeneration. Additionally, it was demonstrated that interstitial macrophage-derived IL-1β induced conversion of *Il1r1*^+^ATII cells into the transitional state via a HIF1α-mediated glycolysis pathway, determined by lineage tracing, single-cell RNA sequencing, and bleomycin mouse model [43]. These findings emphasize the regenerative role of myeloid cells and cytokine-mediated signaling in the ATII cell niche.

## 5. Other Progenitors for Alveolar Repair

### 5.1. Lineage-Negative Epithelial Progenitors/Distal Airway Stem Cells

Aside from ATII cells’ role as progenitors of ATI cells, some cell subpopulations have been identified to repair alveolar damage [80], including basal and club cells. *KRT5^+^*/*TRP63^+^* basal cells in the mouse and human airway epithelium can differentiate into club cells that, in turn, differentiate into goblet and ciliated cells [81,82]. It has been reported that KRT5^+^ basal cell-like cells, derived from SOX2 but not from AXIN2^+^ ATII cells, were present in severely injured alveolar regions induced by an H1N1 influenza infection or bleomycin in mice [15,34,83]. Imai-Matsushima et al. successfully established primary human distal airway epithelial cell culture with feeder cells using growth factors and small molecules to generate alveolar epithelial cells [84]. However, KRT5^+^ progenitors were not observed after infection with a less virulent influenza strain [85]. This elucidates that KRT5^+^ cells are not a significant source for alveolar repair. Moreover, they seemed to have limited capacity to form functional alveolar structures post-infection with the H1N1 influenza virus. These lineage-negative epithelial progenitors (LNEPs) or distal airway stem cells (DASCs) present within the normal distal lung determine alternative pathways for alveolar repair. They require NOTCH signaling to activate the TRP63^+^ and KRT5^+^ basal cell-mediated lung regeneration but later downregulate this pathway to promote ATII cell differentiation [34]. Xi et al. further indicated that hypoxia drives the NOTCH pathway in a HIF1α-dependent manner [33]. *Hif1α* deletion or WNT/β-catenin activity favored murine LNEPs differentiation into ATII cells. In contrast, HIFα was activated in ATII cells and was crucial for their proliferation and spreading via HIF1α-dependent VEGF expression and SDF1/CXCR4 signaling after acute lung injury in mice induced by LPS or hydrochloric acid [86]. ATII cell-specific *Hif1α* knockout mice had a high mortality rate after hydrochloric acid treatment.

### 5.2. Bronchioalveolar Stem Cells

In the murine embryonic lung, multipotent progenitor cells simultaneously express diverse epithelial cell lineages markers, which subsequently become restricted to different cell lineages [87]. Similarly, a transition stage of proSP-C^+^ club cells was observed in the repair after a severe pulmonary injury in mice caused by influenza virus infection or treatment with bleomycin [88]. These murine SCGB1A1^+^proSP-C^+^ bronchioalveolar stem cells (BASCs), located at the BADJ, lost their SCGB1A1^+^ club cell phenotypes and differentiated into ATII cells and ATI cells over time. It has been demonstrated by lineage tracing that BASCs are the major resource for replenishing alveolar epithelium after severe alveolar damage in mice [89]. They were able to regenerate alveolar epithelium in bleomycin-induced lung injury but rarely in lung homeostasis [90,91]. In support, murine club cells are capable of differentiation into alveolar epithelium structure in 3D cultures [92].

Recently, a subpopulation of SCGB1A1^+^ club cells was identified for alveolar repair in a bleomycin-induced lung injury mouse model [93]. These cells express high levels of major histocompatibility complex (MHC) class I and II genes, including H2-K1. They proliferated and differentiated into alveolar lineages, although limited self-renewal and features of cellular senescence were detected. Importantly, these SCGB1A1^+^ H2-K1^high^ club cell-like progenitors can be expanded in vitro and give rise to ATII and ATI cells to rescue denuded alveoli after transplantation into injured mice [93]. This may direct a new therapeutic development. Additionally, Choi et al. identified a distinctive cluster of *Fstl1*^+^*Scgb1a1*^+^ secretory cells with *Id3* and *Porcn* expression and lower levels of some secretory cell markers such as *Scgb3a2* and *Cldn10*, as determined by single-cell RNA sequencing and lineage labeling in mice challenged by bleomycin [94]. These cells also expressed *Cdkn1c*/*p57* involved in cell cycle inhibition, indicating a quiescent characteristic in vivo. Importantly, this cluster may function as the transitional lineage between *Scgb1a1*^+^ secretory cells and *Sftpc*^+^ ATII cells under NOTCH inhibition by treatment with a γ-secretase inhibitor DAPT or *Rbpj* knockdown in murine airway organoids. By contrast, secretory cells isolated from mice with constitutive *Notch1* activation retained only airway cells. A substantial increase of ATII cells derived from the secretory cells was observed during bleomycin-induced lung repair in mice carrying a conditional dominant-negative mutant of mastermind-like 1 inhibiting NICD-induced NOTCH signaling. Of note, NOTCH signaling is regulated by IL-1β, which is required for ATII cell formation during bleomycin-induced alveolar regeneration in mice [94]. In addition, the transcription factor FOSL2/FRA-2 is essential for secretory cell fate conversion to ATII cells with distinct genetic and epigenetic signatures of secretory lineages, determined by ATAC-seq analysis, knocked down experiments, and organoid cultures. Consistent with mice studies, KDR/FLK-1^+^ secretory cells from human distal lungs can convert into ATII cells under NOTCH inhibition by DAPT treatment in organoids [94]. A subset of UPK3A^+^ club cells, located near neuroendocrine bodies and the BADJ, has also been shown to generate ATII and ATI cells in bleomycin-induced lung injury in mice [95].

Unexpectedly, lineage-labeled HOPX^+^ ATI cells proliferate and generate ATII cells after pneumonectomy [96]. In addition to ATII cell-generating organoids, individual HOPX^+^ ATI cells form 3D organoids composed of ATI and ATII cells ex vivo. Inhibition of TGF-β signaling markedly increases the colony-forming efficiency of organoids. In particular, HOPX^+^IGFBP2^+^ ATI cells represent the terminally differentiated population, distinguishing the heterogeneity of ATI cells and cell fate during lung regeneration [97]. These studies surprisingly bring out ATI cell plasticity after lung injury. Increasing reports on progenitor cell identity indicate the complexity of cells involved in alveolar repair, depending on different types or levels of lung injury. The mechanisms and microenvironment underlying alveolar re-epithelialization are still elusive. Regulation of ATII cell proliferation and differentiation requires more studies.

## 6. Therapeutic Targets in Emphysema

### 6.1. Therapeutics and Emerging Treatments

There is no cure for emphysema, but appropriate therapeutic strategies help manage symptoms and prevent exacerbations [98]. Pharmacological treatments for emphysema involve inhaled steroids, bronchodilators, and mucolytics [99]. Thus far, they are less effective in this disease than in non-emphysematous COPD patients. This indicates the urgent need for novel targets or approaches focused on the cellular processes of alleviating chronic lung injury and regenerating the lungs. New therapeutic drugs include phosphodiesterase-4 (PDE4) inhibitors with bronchodilator and anti-inflammatory properties. Roflumilast is approved to treat COPD and asthma [100]. PDE4 is mainly expressed in immune cells and many cell types in the lung, including ATII cells. It controls the intracellular second messenger cyclic adenosine monophosphate (cAMP) by promoting its hydrolysis into an inactive metabolite, thus affecting numerous physiological processes, including cell proliferation and differentiation. Additionally, the expression of surfactant proteins is regulated by cAMP [101]. Therefore, inhibition of PDE4 can increase cAMP levels and activate downstream signaling for immune balance and improved alveolar function. This implies its efficacy in promoting alveolar regeneration. Other PDE inhibitors are in clinical trials.

Furthermore, characteristics of cellular senescence were observed in emphysema patients [46]. Treatments targeting cellular senescence have beneficial effects on animal models of emphysema. Melatonin, an indoleamine hormone involved in regulating the circadian rhythm and cell growth, is reduced in COPD patients [102]. Its intraperitoneal injection in rats challenged with cigarette smoke and LPS decreased endoplasmic reticulum (ER) stress, bronchial and alveolar epithelial cell apoptosis, and emphysema development via upregulating anti-aging protein SIRT-1 expression in the lungs [30]. It has also been shown that *p16* deletion decreased murine ATII cell senescence and the pathophysiological changes in alveoli induced by cigarette smoke exposure [103], suggesting the contribution of cell cycle regulators to alveolar structure homeostasis. Senotherapy can be an interesting target to rejuvenate cells for lung regeneration. Several mitochondria-targeted drugs are in clinical trials for other aging-related diseases.

### 6.2. Natural Supplementations

Natural supplements such as dark vegetables, vitamin K [104], copper [105], and hyaluronan [106] affect lung function. Accumulating evidence shows the benefits of quercetin, a natural flavonoid with cytoprotective, anti-inflammatory, and immune regulatory properties, in COPD patients, cardiovascular problems, and respiratory infections [107,108]. Additionally, vitamins affect lung physiology and thus may prevent respiratory diseases. Vitamin A deficiency was linked to emphysema severity, and its supplementation protected against this disease development [104,109]. All-trans retinoic acid (ATRA), an active metabolite of vitamin A, has disease-modifying effects that promoted alveolar epithelium regeneration in rats with elastase-induced emphysema [110]. It reduced inflammation via ERK and JAK-STAT pathways, elevated antioxidant activity in the lung, and decreased protease-antiprotease imbalance in BALF. Recently, there has been an increasing interest in vitamin D supplementation since its deficiency in serum was associated with emphysema progression in men [111] and mice [112]. Ghosh et al. also found a correlation between vitamin D deficiency and decreased lung function in smokers; controversially, there was no significant correlation with CT measures of emphysema [113]. Inhalation of 1,25-dihydroxy vitamin D3 revealed an alveolus-regenerating effect and improved respiratory function in mice with elastase-induced emphysema [114], suggesting its potent activity in lung repair. More research is warranted to discover specific nutrients that can stimulate the repair of damaged alveoli.

### 6.3. Cell Therapies

Stem and progenitor cells are widely considered promising therapies for respiratory diseases [115]. Their capability to differentiate into various cell types, strong immunosuppressive and anti-apoptotic properties, and paracrine effects might help regenerate the lung after injury. Different stem cells and delivery routes have been used in mice with elastase-induced emphysema [115]. The differentiation ability of mesenchymal stem cells (MSCs) into ATII cells was studied in vitro [116]. In addition, human pluripotent stem cell (hPSC)-derived alveolar epithelial cells phenotypically and functionally resemble primary ATII cells [117]. Cell transplantation of these ATII cells at a dose of 3 × 10^6^ cells/animal improved alveolar structure in rat lung injury induced by bleomycin [118]. This provides potential alternatives for therapeutic approaches.

Cell therapies have been used in clinical trials, although to our knowledge there is a lack of ATII cell therapy in emphysema. Autologous stem cells from bone marrow, peripheral blood, and adipose tissue are the most commonly used. Patients with advanced emphysema (GOLD4) received the autologous infusion of bone marrow mononuclear cells and were followed up for 3 years by spirometry [119]. Their monitoring showed the safety of cell therapy without significant adverse effects. The improvement in lung function was also observed in this group compared to a placebo-controlled randomized trial of MSCs. Indeed, intravenous MSC infusion reached the lung within 30 min [120]. However, it had a lower retention rate in the emphysematous lung compared to normal lung, which was correlated to low FEV1 and DLCO (diffusing capacity of the lung for carbon monoxide). Although this study did not find statistically significant improvement in spirometry, MSC infusion reduced oxidative stress and inflammatory response. Peripheral blood mononuclear cells (PBMCs) isolated from patients had significantly reduced levels of inflammatory genes, including *IL-8*, *NFKB1*, *TLR2*, *CD44*, *STAT3*, and increased metabolic genes between 1 h and 2 days after infusion, as detected by RNA sequencing [121]. The number of these differentially expressed genes declined by 7 days. More specifically, culturing PBMCs with MSC-conditioned medium or post-MSC infusion plasma reduced IL-8 and IL-1β production, suggesting the presence of soluble anti-inflammatory factors in plasma and paracrine regulatory mechanisms. Soluble TNF receptor-1 (sTNFR1) and TGF-β levels and microRNAs that target inflammatory genes were further detected in the plasma after infusion. A combination of one-way endobronchial valve insertion and allogeneic bone marrow-derived MSCs administration was also applied in patients with advanced emphysema without direct adverse events [122]. The circulating C-reactive protein level was reduced after treatment with MSCs, and elevated quality of life was reported.

While regenerative medicine in lung diseases shows encouraging results, many challenges and questions remain. Additional in vivo studies on the alveolar epithelial function and clinical evidence using novel treatments, long-term evaluations, and mechanistic studies are still needed. It may be beneficial to stimulate lung regeneration by increasing the progenitor pool’s ability to proliferate and decreasing alveolar wall destruction. Nevertheless, still little is known regarding the regulation of ATII cell function and re-epithelialization after lung injury. Besides, the regenerative ability of ATII cells needs to be further evaluated for targeting respiratory diseases.

## 7. Conclusions

The lung is a delicate organ made of interdependent tissue [80]. Its main role is a gas exchange that depends on functional alveoli. Mechanical forces, intercellular interactions, transcription factors, mediators, and signaling pathways orchestrate complex lung physiology and homeostasis. Specifically, ATII cells devote to the alveolar microenvironment and innate immunity. Importantly, they have stem cell activity for self-renewal, and their differentiation into ATI cells contributes to re-epithelialization after injury [3,15]. Different signaling pathways precisely regulate ATII cell proliferation, transitional state, and differentiation. ATII cell niches, including mesenchyme, immune cells, and pulmonary vasculature, institute a specific microenvironment to support their function. Moreover, other progenitor cells for alveolar repair and cell lineages that can gain stem cell properties have been discovered by lineage tracing approaches and lung injury models in animals [33,88].

Increasing evidence shows multiple dysregulated signaling pathways in ATII cells in emphysema [25,28,46], contributing to apoptosis/proliferation imbalance and alveolar destruction. Besides, less responsive mesenchymal cells [8] and dysfunctional endothelium [63] were observed in patients. Therefore, targeting ATII cell proliferation/differentiation and niches for re-epithelialization might rescue alveolar wall damage. Regenerative medicine is an exciting approach to pulmonary diseases. Stem cell therapies in animals and humans with emphysema show the treatments’ safety and promising results, although they are still at an early stage. Treatments targeting pulmonary vasculature, cellular senescence, and natural supplementations also gain attention to lung repair. Nevertheless, current pharmacotherapy is still insufficient in treating emphysema. More evidence is urgently needed to fill the gap in our knowledge of regulating ATII cell function and initiating lung regeneration for novel treatments for emphysema and other lung diseases.

## Figures and Tables

**Figure 1 cells-11-02055-f001:**
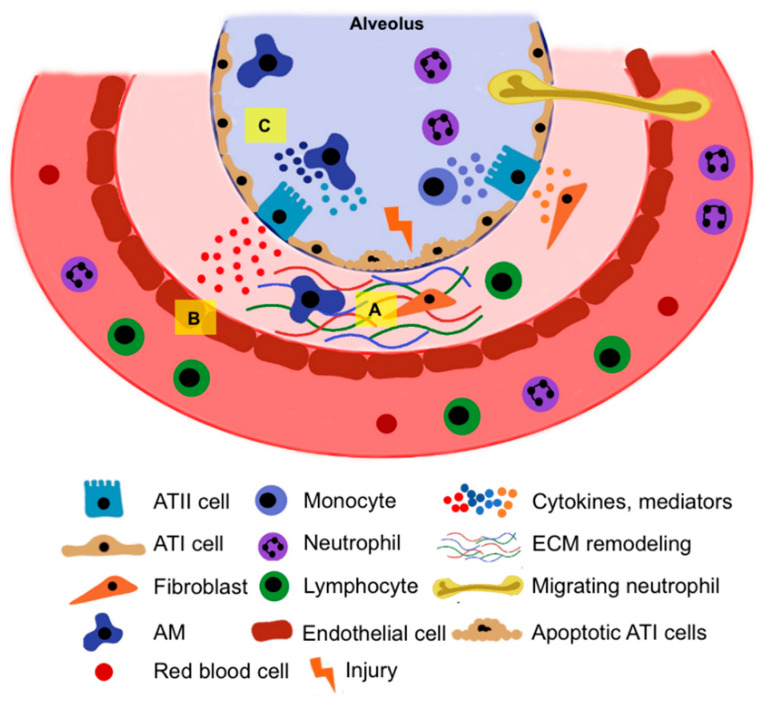
Various cell types are part of the ATII cell niche, providing a specific microenvironment to support alveolar re-epithelialization after injury. (**A**) Mesenchyme, including fibroblasts, extracellular matrix (ECM), and pericytes around endothelium and alveoli, can sense microenvironmental changes and mechanical stress and interact with alveolar epithelium, contributing to tissue remodeling. (**B**) Activated endothelial cells proliferate and regulate alveolar epithelial cell repair after injury to maintain alveolar–capillary function and integrity. (**C**) Immune cells are a component of the regenerative alveolar type II (ATII) cell niche. Alveolar macrophages (AM) are important in innate immunity, regulating adaptive immunity, and recruiting neutrophils and monocytes into the lungs. They defend against pathogens while supporting ATII cell proliferation and differentiation to alveolar type I (ATI) cells. Together, various reciprocal interactions, transcription factors, mediators, and signaling pathways among different cell types orchestrate complex lung repair and homeostasis.

**Figure 2 cells-11-02055-f002:**
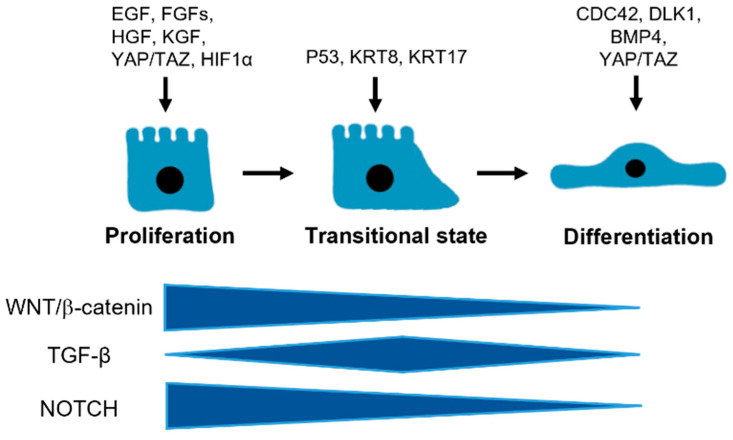
The regulation of ATII cell proliferation and differentiation. WNT/β-catenin, TGF-β, and NOTCH pathways dynamically regulate ATII cell proliferation and differentiation into ATI cells, although there are still many unknowns during the transitional state. Other transcription factors, growth factors, and signaling molecules are also involved.

**Figure 3 cells-11-02055-f003:**
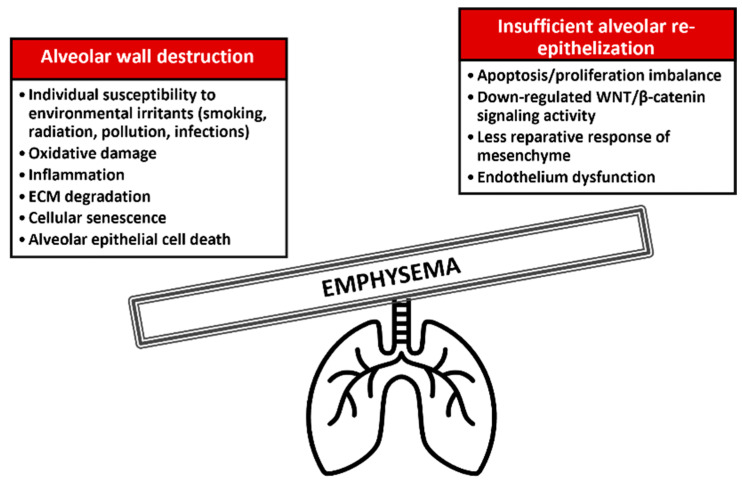
Alveolar wall destruction and impaired re-epithelialization in pulmonary emphysema.

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
