# Peer review of "Impaired Alveolar Re-Epithelialization in Pulmonary Emphysema"

_cells, 2022, doi:10.3390/cells11132055_

Round 1

Reviewer 1 Report

This manuscript by Lin et al. is a very nicely written review summarizing and discussing the current understanding of ATII cells in the pathophysiology of COPD/emphysema. Overall, it is a very clear review, however sometimes I experience that the authors bring in too much information about other cell types making the review a bit long and unspecific. I would suggest removing section 7.1, 7.1 and the 2nd part of 7.3, which hopefully will result in a more AT-specific review. It would also be great if the authors could add a figure summarizing all different pathways discussed in 3.1-3.6.

Reviewer 2 Report

Thank you very much for giving me the opportunity to review the manuscript entitled “Dysfunctional alveolar re-epithelialization in pulmonary emphysema”.

General comment

In this review article the authors provide a thorough overview regarding our current understanding of the renewal of the alveolar epithelium in the lung. Based on (but not limited to) the recent literature they introduce the relevant mediators/ factors involved in controlling alveolar epithelial type II (AE2 cell) proliferation and differentiation to alveolar epithelial type I cells via a transitional stage. In addition, the authors give a detailed description of the cells necessary for creating the stem cell niche within fine lung parenchyma and also introduce other stem cells which have the potential to contribute to regeneration of the alveolar epithelial lining such as basal cells or bronchio-alveolar stem cells. In general, the authors succeed in building a holistic concept of our current knowledge of the regeneration of the alveolar epithelial lining. However, after having read the manuscript several times I was asking myself whether the authors addressed the topic as given by the title appropriately. From my perspective, the title and the text are not very well harmonized and as a reader one might be disappointed if one expects to get information on “dysfunctional alveolar re-epithelialization in pulmonary emphysema”.  Most of the referred literature is based on investigations in animal models of lung injury and fibrosis or disturbed alveologenesis. The authors failed to link the concepts of disturbed alveolar epithelial regeneration given e.g. in section 3 to the pathophysiology/ pathology of emphysema in section 6. After all, the role of dysfunctional alveolar re-epithelialization in in the pathophysiology of emphysema remains quite speculative.  Also, I think that this review is not focused enough; e.g. section 7 dealing with “Therapeutic targets in emphysema” and can be dramatically shortened since hardly any of the discussed therapeutic approaches really addresses the alveolar epithelial regeneration. Based on these more general comments I have several more specific comments.

  1. Abstract lines 17-18: The factors mentioned in line 15 were described to be dysregulated in emphysema. However, their roles in the destructive process resulting in pulmonary emphysema is not well defined, e.g., by functional or imaging studies showing abnormalities of the alveolar epithelium (e.g. denuded basement membranes, intermediate cells, hyperplasie of alveolar epithelial type II cells … ). I think the sentence “However, their dysregulation was observed in patients with pulmonary emphysema, which is characterized by alveolar wall destruction and impaired gas exchange.” is misleading as it implies that impaired alveolar epithelial regeneration results in destruction which has not been proven in the context of pulmonary emphysema by mechanistic studies.
  2. Introduction: I would suggest to start the introduction with a definition of pulmonary emphysema.
  3. Introduction, lines 30-32: related to comment 1 – the statement that alveolar epithelial cell death and impaired re-epithelialisation results in parenchymal destruction in the context of pulmonary emphysema is not proven and speculative based on the provided references. Also reference 2 speculates about the role of alveolar epithelial type II cells in emphysema but is more conservative in the interpretation of the available knowledge: e.g. it says in ref 2: “In addition to the potential role of inflammatory cells, notably of neutrophils (41, 42) and macrophages (7, 43), the roles of alveolar type II cells and capillary endothelial cells in driving the process of destruction have remained unclear”.
  4. Introduction, lines 70-72: in healthy lungs vascular smooth muscle cells (VSMS) are not part of the inter-alveolar septa in most regions of the lung so that they are not covered by alveolar epithelial cells as stated here. In human lungs there are VSMC located in the entrance rings of alveoli in the first generation of intra-acinar conducting airways – this is true. But this represents only a very minor fraction of the gas-exchanging region.
  5. Section 3. In think the authors provide here a very good overview of the concepts of alveolar epithelial regeneration in general. However, observations are largely based on animal models of lung injury and fibrosis or lung development. The authors should try to connect these findings better to the pathophysiology of pulmonary emphysema. Since many references originate from investigations of lung injury and fibrosis the authors should try to answer the following question: What is the difference in the dysregulation of alveolar epithelial regeneration in emphysema and pulmonary fibrosis? Or why do lungs develop pulmonary fibrosis in presence of impaired alveolar epithelium and not emphysema (as in Ref. 27)
  6. Also, section 3 would benefit from a brief summary of the dynamics and the extend of the cell turnover in the alveolar epithelial lining.
  7. Section 3.1. How could abnormal mechanical forces in pulmonary emphysema result in abnormal alveolar epithelial regeneration and potentially disease progression? The authors provide important insight into the role of mechanical forces in the context of lung development and pulmonary fibrosis. However, the link to emphysema is missing!
  8. Lines 114-115: Tight junctions have been suggested to be involved in mechano-transduction but the authors should also mention here the role of adherence junctions
  9. Lines 447-449: How can alveolar re-epithelialisation repair lost alveoli? This is a quite simplistic view which does not reflect the reality in regenerative medicine. It needs more than epithelial cells to build new interalveolar septa – e.g. the scaffold and the alveolar capillary network which is really complex, fibroblasts ..!
  10. Section 6: The molecular markers suggest that AE2 cells are dysfunctional in terms of the regeneration of the alveolar epithelial lining. But I am missing here a detailed presentation of data from functional studies demonstrating that impaired regeneration of the alveolar epithelium results in a destruction of interalveolar septa combined a reduction in the surface area for gas exchange and with an irreversible enlargement of distal airspaces. I think this is crucial having the title of the paper in mind and would address the expectations of the readership more appropriate. The authors should also be careful to distinguish between studies on lung development and pulmonary emphysema. E.g. ref. 83 is not really addressing pulmonary emphysema – according to the study design Ref. 83 investigates mechanisms of impaired lung development (let’s say broncho-pulmonary dysplasia).
  11. Lines 453-454: is there any evidence from imaging studies that there is a denudation of the alveolar epithelial basal lamina/membrane (or denuded epithelium as the authors call it)? I think this would be a strong argument in favour of the functional relevance of an impaired regeneration capacity of the AE2 cells. At a morphological level An impaired regeneration could not only manifest as denuded basal lamina/ membrane but also the existence of intermediate cells (e.g. epithelial cells sharing features of both AE1 and AE2 cells).
  12. Section 7: I think this part should be shortened to focus this manuscript on its main topic, the regeneration of the alveolar epithelial lining.

Round 2

Reviewer 2 Report

The authors addressed my comments appropriately. Thank you very much!